# Stress Hyperglycemia and Osteocalcin in COVID-19 Critically Ill Patients on Artificial Nutrition

**DOI:** 10.3390/nu13093010

**Published:** 2021-08-28

**Authors:** Francisco Arrieta, Victoria Martinez-Vaello, Nuria Bengoa, Marta Rosillo, Angélica de Pablo, Cristina Voguel, Rosario Pintor, Amaya Belanger-Quintana, Raquel Mateo-Lobo, Angel Candela, José I. Botella-Carretero

**Affiliations:** 1Department of Endocrinology and Nutrition, Hospital Universitario Ramón y Cajal, 28034 Madrid, Spain; arri68@hotmail.com (F.A.); vmartinezvaello@gmail.com (V.M.-V.); nuria.bengoa@salud.madrid.org (N.B.); raquel.mateo@salud.madrid.org (R.M.-L.); 2Centro de Investigación Biomédica en Red Fisiopatología de la Obesidad y Nutrición (CIBERobn) & Instituto Ramón y Cajal de Investigación Sanitaria (IRyCIS), 28034 Madrid, Spain; 3Department of Biochemistry, Hospital Universitario Ramón y Cajal, 28034 Madrid, Spain; marta.rosillo@salud.madrid.org; 4Department of Anesthesiology, Hospital Universitario Ramón y Cajal, 28034 Madrid, Spain; angelicadepabl@hotmail.com (A.d.P.); crivomed@gmail.com (C.V.); angel.candela@salud.madrid.org (A.C.); 5Department of Pharmacy, Hospital Universitario Ramón y Cajal, 28034 Madrid, Spain; mariarosario.pintor@salud.madrid.org; 6Department of Pediatrics, Hospital Universitario Ramón y Cajal, 28034 Madrid, Spain; amaya.belanger@salud.madrid.org

**Keywords:** COVID-19, hyperglycemia, parenteral nutrition, enteral nutrition, osteocalcin

## Abstract

We aimed to study the possible association of stress hyperglycemia in COVID-19 critically ill patients with prognosis, artificial nutrition, circulating osteocalcin, and other serum markers of inflammation and compare them with non-COVID-19 patients. Fifty-two critical patients at the intensive care unit (ICU), 26 with COVID-19 and 26 non-COVID-19, were included. Glycemic control, delivery of artificial nutrition, serum osteocalcin, total and ICU stays, and mortality were recorded. Patients with COVID-19 had higher ICU stays, were on artificial nutrition for longer (*p* = 0.004), and needed more frequently insulin infusion therapy (*p* = 0.022) to control stress hyperglycemia. The need for insulin infusion therapy was associated with higher energy (*p* = 0.001) and glucose delivered through artificial nutrition (*p* = 0.040). Those patients with stress hyperglycemia showed higher ICU stays (23 ± 17 vs. 11 ± 13 days, *p* = 0.007). Serum osteocalcin was a good marker for hyperglycemia, as it inversely correlated with glycemia at admission in the ICU (*r* = −0.476, *p* = 0.001) and at days 2 (*r* = −0.409, *p* = 0.007) and 3 (*r* = −0.351, *p* = 0.049). In conclusion, hyperglycemia in critically ill COVID-19 patients was associated with longer ICU stays. Low circulating osteocalcin was a good marker for stress hyperglycemia.

## 1. Introduction

Severe acute respiratory syndrome coronavirus 2 (SARS-CoV-2) is the cause of the coronavirus disease 2019 (COVID-19) pandemic, which has affected more than two hundred million individuals and is the cause of more than four million deaths worldwide as of this writing (https://coronavirus.jhu.edu/map.html, last accessed on 24 August 2021). Hyperglycemia is a risk factor for a more severe course of the disease, as hospitalized COVID-19 patients with diabetes show longer hospital stays than patients without diabetes [1]. They suffer disproportionately from acute COVID-19, with higher rates of serious complications and death [2]. Chronic inflammation, increased coagulation activity, immune response impairment, and potential direct pancreatic damage by SARS-CoV-2 might be among the underlying mechanisms for this association [3]. Furthermore, an important proportion of COVID-19 patients need admission to an intensive care unit (ICU) and may develop stress hyperglycemia—whether or not having a previous history of diabetes mellitus—which has been shown to be a prognostic factor [4,5].

Among the factors associated with stress hyperglycemia, the need for parenteral nutrition (PN)—which is normally part of the nutritional therapy together with enteral nutrition (EN) in critically ill patients [6]—might be one of them, especially in older individuals [7]. Besides, several interactions between some counter-regulatory hormones, adipokines, and inflammatory cytokines produce excessive production of glucose by the liver and insulin resistance at the peripheral tissues [8]. The resultant hyperglycemia further exacerbates the inflammatory and oxidative stress response, potentially setting up a vicious cycle whereby hyperglycemia leads to further hyperglycemia [9].

In the last few years, circulating osteocalcin, an osteoblast-specific protein, has shown extraskeletal metabolic activity, such as promoting insulin secretion and increasing peripheral insulin sensitivity [10]. Reduced circulating osteocalcin was found in patients with type 2 diabetes mellitus and also associated with insulin sensitivity [11,12].

Therefore, given this previous knowledge, we aimed to study the possible association of stress hyperglycemia in COVID-19 critically ill patients with a worse prognosis compared to non-COVID-19 patients. We also aimed to explore the possible associations of hyperglycemia with circulating osteocalcin concentrations, the composition of artificial nutrition, and serum markers of inflammation.

## 2. Materials and Methods

### 2.1. Subjects and Measurements

Fifty-two patients were included in this study: 26 consecutive patients with severe COVID-19 requiring admission to the ICU and 26 non-COVID-19 critically ill postsurgical patients. The latter were historical controls before the COVID-19 pandemic occurred and had the following procedures: 7 cardiovascular surgery, 12 renal/urological surgery, 6 pelvic surgery, and 1 surgery for traumatic hemothorax. Patients with COVID-19 were diagnosed by the presence of SARS-CoV-2 in respiratory specimens by real-time polymerase chain reaction (RT-PCR) in pharyngeal swabs. All COVID-19 patients were critically-ill and on mechanical ventilation. They received our standard treatment protocol, including low molecular-weight heparin, glucocorticoids, and tocilizumab. Non-COVID-19 patients were all complicated postsurgical ones in need of mechanical ventilation and vasoactive drugs. Patients under 18 y were excluded as well as those with active cancer, pregnancy, or with end-stage renal or liver disease before the diagnosis of COVID-19.

Patients in both groups were categorized as with or without diabetes mellitus based on a previous diagnosis. Stress hyperglycemia was defined as a plasma glucose level of ≥140 mg/dl. Continuous insulin infusion (50 IU of Actrapid in 50 mL of 0.9% saline using an IV pump) was started when blood glucose was ≥180 mg/dl and adjusted for a glycemic range of 140–179 mg/dl [13]. Capillary glycemia was measured every one or two hours to adjust the infusion rate. When blood glucose fell to <140 mg/dl, insulin infusion was stopped and subcutaneous insulin started. Capillary glycemia was then measured every six hours. ICU stay and total hospital stay were recorded as well as mortality.

The type of composition of artificial nutrition administered to the included patients were recorded. PN was delivered through a central line as soon as the patient was hemodynamically stable. Individualized formulae or standardized commercial bags were prepared by the hospital pharmacy. We aimed at 20–25 kcal/Kg/day, with a proportion of 3–6 g/Kg/day for glucose, 1.0 g/Kg/day for amino-acids, and less than 1 g/Kg/day for lipids, with 7–10 g/day of essential fatty acids. Vitamins and trace elements were also added by the hospital pharmacy. For those patients with ICU stays above 7 days, and whenever possible, EN was started with a standard, fiber-free formulation and PN gradually tapered as EN was tolerated and increased.

### 2.2. Ethics

The study protocol was approved by the Institutional Ethics Committee of our center (study code 147/20) and performed according to the Declaration of Helsinki. Written or verbal informed consent was obtained from all patients.

### 2.3. Analytical Assays

Serum concentrations of glucose and other biochemical variables were measured with an Architect c16000/i2000-analyzer (Abbot Diagnostics, Maidenhead, UK) and HbA1c by high-performance liquid chromatography (A. Menarini Diagnostics, Bagno a Ripoli, Italy). Immunoanalysis was employed for the measurement of C-reactive protein (CRP), procalcitonin (Abbott, Illinois, IL, USA) and D-dimer (Siemens, Münich, Germany), and interleukins 6 (IL-6) and 12 (IL-12) by enzyme-linked immunosorbent assay (Invitrogen, Waltham, MA, USA) at ICU admission. Serum osteocalcin was measured by electrochemiluminescence Cobas-e601 (Roche Diagnostics, Rotkreuz ZG, Switzerland), with normal range of 15.0–45.0 µg/L also at ICU admission. The intra- and inter-assay coefficients of variation were below 10%.

### 2.4. Statistical Analysis

We used GRANMO 7.12 [14] for sample size analysis. The primary outcome was to study the possible association of stress hyperglycemia in COVID-19 critically ill patients with a worse prognosis compared to non-COVID-19 patients. In order to find a mean difference of 10 days in ICU stay with a SD of 10, we needed a sample size of at least 16 individuals in each group for a two-tail estimates setting α at 0.05 and β at 0.20. Secondary outcomes were the possible associations of hyperglycemia with circulating osteocalcin concentrations, the composition of artificial nutrition, and serum markers of inflammation. Based on our previous results [12], in order to find a mean difference of 7.6 µg/L in osteocalcin concentrations with a SD of 8.8, we needed a sample size of at least 21 individuals in each group for a two-tail estimates setting α at 0.05 and β at 0.20. Sample size analyses for the composition of artificial nutrition and inflammatory markers were not performed.

Results are expressed as means ± SD unless otherwise stated. The Kolmogorov–Smirnov statistic was applied to continuous variables. Logarithmic or square root transformations were used as needed to ensure normal distribution of the variables. To compare discontinuous variables, we used the χ^2^ test and Fisher’s exact test as appropriate. Unpaired *t*-tests or Mann–Whitney U tests were used to compare the central tendencies of the different groups as appropriate. Bivariate correlation was employed to study the association between two continuous variables using Pearson or Spearman’s tests as appropriate. Two-way ANOVA was employed to analyze the effect of both COVID-19 and stress hyperglycemia on the studied continuous prognostic variables and corrected χ^2^ test for categorical ones. Finally a multivariate linear regression analysis was also performed with a backwards strategy. Analyses were performed using SPSS 17 (SPSS Inc, Chicago, IL, USA). *p* < 0.05 was considered statistically significant.

## 3. Results

### 3.1. Baseline Characteristics of Patients

Of the 52 initially included patients, the results of 49 patients were finally analyzed, as three patients with COVID-19 had undetectable levels of osteocalcin (Table 1). Patients with COVID-19 were younger, with a higher proportion of males. Previous diagnosis of diabetes mellitus was similar in both groups as well as their HbA1c.

We did not find a difference in glycemia at ICU admission in the proportion of patients with stress hyperglycemia or in mean glycemia at ICU. However, patients with COVID-19 more frequently needed insulin infusion therapy for glycemic control and showed lower osteocalcin concentrations. Patients with COVID-19 were also on artificial nutrition for longer (both PN and EN) and received higher energy per day than non-COVID-19 ones but with similar amounts of delivered glucose per day. Serum CRP and procalcitonin at ICU admission were similar in both groups (Table 1).

Mortality did not differ between groups. Non-COVID-19 patients died as a result of cardiac arrest (*n* = 3), pulmonary embolism (*n* = 1), and multiorgan failure after septic shock (*n* = 3). COVID-19 patients died as a result of respiratory distress syndrome (*n* = 6) and pulmonary embolism (*n* = 2). The possibility of stroke and or any other CNS complications as the cause of death could not be accurately assessed due to profound sedation in these patients (Table 1).

### 3.2. Impact of Stress Hyperglycemia

When both patients with and without COVID-19 were considered together and classified according to the presence of stress hyperglycemia, the latter group showed a higher proportion of previous diagnosis of type 2 diabetes mellitus and higher HbA1c levels (Table 2). They were on PN for longer, received higher amounts of energy and glucose from artificial nutrition, and showed higher ICU stays.

Total hospital stay did not correlate with glucose control, but ICU stay correlated with glycemia at admission in ICU (*r* = 0.337, *p* = 0.018) and at day 2 (*r* = 0.427, *p* = 0.015). Those patients with stress hyperglycemia showed higher ICU stays. As expected, those patients with longer ICU stays needed more days on artificial nutrition for both PN (*r* = 0.741, *p* < 0.001) and EN (*r* = 0.829, *p* < 0.001).

### 3.3. Circulating Osteocalcin as a Marker for Hyperglycemia and Prognosis

Circulating osteocalcin was lower in patients with COVID-19 (Table 1). It was lower in patients with stress hyperglycemia (Table 2) and in those in need for insulin infusion therapy (7.0 ± 4.7 vs. 12.3 ± 7.0 µg/L for subcutaneous insulin, *t* = 3.672, *p* = 0.001).

Circulating osteocalcin concentrations inversely correlated with glycemia at the day of admission in the ICU (*r* = −0.476, *p* = 0.001) and at days 2 (*r* = −0.409, *p* = 0.007) and 3 (*r* = −0.351, *p* = 0.049) (Figure 1). Osteocalcin did not correlate with HbA1c (*r* = −0.207, *p* = 0.225) or age (*r* = 0.145, *p* = 0.319), and their levels were similar in men and women (*t* = 0.482, *p* = 0.632).

### 3.4. Impact of the Composition of Artificial Nutrition

The amount of energy delivered through PN did not correlate with glycemia at the day of admission in the ICU (*r* = 0.245, *p* = 0.097), at day 2 (*r* = 0.211, *p* = 0.245), or 3 (*r* = 0.069, *p* = 0.750). Conversely, the amount of glucose delivered through PN was positively correlated with glycemia at the day of admission in the ICU (*r* = 0.395, *p* = 0.006). In addition, mean glucose and energy delivered through artificial nutrition correlated with mean glycemia during ICU stay (*r* = 0.399, *p* = 0.007 and *r* = 0.291, *p* = 0.048, respectively) (Figure 2).

Those patients in need of insulin infusion therapy compared with those on subcutaneous insulin received higher energy (1221 ± 256 vs. 923 ± 294 kcal/day, respectively, *t* = 3.709, *p* = 0.001) and glucose delivered through artificial nutrition (146 ± 22 vs. 133 ± 22 g/day, respectively, *t* = 2.112, *p* = 0.040).

### 3.5. Circulating Inflammatory Markers

Glycemia at the day of admission in the ICU was positively correlated with IL-12 (*r* = 0.454, *p* = 0.038) but not with CRP (*r* = 0.065, *p* = 0.722), procalcitonin (*r* = 0.145, *p* = 0.446), IL-6 (*r* = 0.138, *p* = 0.551), or D-dimer (*r* = 0.232, *p* = 0.312) (Figure 3).

Patients with stress hyperglycemia had more similar inflammatory markers at ICU admission than those without it (Table 2). The same occurred between those patients in need for insulin infusion therapy compared with those on subcutaneous insulin (*p* > 0.05 for IL-12, CRP, procalcitonin, D-dimer, and IL-6).

### 3.6. Ancillary Analyses

In order to correct for the effects of both the presence of COVID-19 and stress hyperglycemia in prognostic parameters and osteocalcin, we performed a two-way analysis of variance for continuous variables and corrected χ^2^ test for categorical ones. Separated data in four groups are shown in Table 3.

ICU stay was higher with both COVID-19 diagnosis and the presence of stress hyperglycemia, indicating these two were independent prognostic factors. Conversely total hospital stay and mortality were not associated with either COVID-19 diagnosis or the presence of stress hyperglycemia. Circulating osteocalcin was lower in both COVID-19 patients and with the presence of stress hyperglycemia, but CRP and procalcitonin showed no associations (Table 3).

Insulin infusion was more frequent for both COVID-19 diagnosis and the presence of stress hyperglycemia. The time on artificial nutrition, both on TPN and EN, was longer in those patients with COVID-19 and with stress hyperglycemia. The amount of energy and glucose delivered by artificial nutrition were also associated with both the presence of COVID-19 and stress hyperglycemia and with their interaction (Table 3).

Finally, a multivariate linear regression analysis was also performed with a backwards strategy, introducing COVID-19 diagnosis, the presence of stress hyperglycemia, the need for insulin infusion, age, and sex as independent variables and ICU stay as the dependent variable. Both COVID-19 diagnosis (β = 0.488, *p* < 0.001) and the presence of stress hyperglycemia (β = 0.394, *p* = 0.001) were retained by the model (R^2^ = 0.383, F = 14.297, *p* < 0.001). When osteocalcin was introduced as the dependent variable, both COVID-19 diagnosis (β = −0.375, *p* = 0.005) and the presence of stress hyperglycemia (β = −0.344, *p* = 0.009) were retained by the model (R^2^ = 0.347, F = 12.224, *p* < 0.001).

Another multivariate regression analysis model was performed as to take into account the artificial nutrition characteristics on ICU stay, which was introduced as the dependent variable. The independent variables in this model were the days on artificial nutrition, both TPN and EN, the amount of energy and glucose delivered by artificial nutrition, and the need for insulin infusion. The presence of COVID-19 and stress hyperglycemia could not be introduced in the model, as they showed collinearity with the variables of artificial nutrition. The variables retained by the model (R^2^ = 0.965, F = 438.5, *p* < 0.001) were the days on both TPN (β = 0.535, *p* < 0.001) and EN (β = 0.652, *p* < 0.001) and the need for insulin infusion (β = 0.066, *p* = 0.031).

## 4. Discussion

In the present study, we have shown that hyperglycemia in critically ill COVID-19 patients was associated with longer ICU stays and higher amounts of glucose delivered through artificial nutrition. Low circulating osteocalcin was lower in COVID-19 patients, in those with stress hyperglycemia, and in those in need for insulin infusion therapy. Therefore, osteocalcin could be considered a useful marker for stress hyperglycemia and prognosis at ICU.

Osteocalcin, while playing important roles in bone remodeling, also contributes to glucose metabolism by affecting both insulin secretion and insulin sensitivity [15]. In vitro, co-cultures of pancreatic islets and wild-type osteoblasts stimulated insulin secretion, whereas knockout osteocalcin osteoblasts did not [15]. Furthermore, the link between bone and glucose metabolism is supported by clinical observations indicating that patients with diabetes show an increased risk of fractures because of osteopenia or osteoporosis [16,17] and similarly in animal models [10].

Circulating osteocalcin was reduced in patients with severe COVID-19, in accordance with a recent report in which 40 patients were compared with 57 non-COVID-19 controls in a cross-sectional design [18]. We further analyzed the serum osteocalcin association with glycemia in COVID-19 patients, as we and other authors reported in the past this relationship in other type of patients [12,19]. We found that circulating osteocalcin was inversely correlated with glycemia: it was lower in those with stress hyperglycemia, those in need for insulin infusion therapy, and also associated with longer ICU stays. To our knowledge, this is the first time that this association is reported in critically ill patients.

Diabetes mellitus has shown to be a risk factor for a worse prognosis in patients with COVID-19 [20,21] as well as those who develop stress hyperglycemia [4]. Furthermore, the severity of COVID-19 illness increases progressively in relation to glucose abnormalities at admission [22], and this has also been shown to happen in patients without a previous diagnosis of diabetes [23]. Conversely, a recent report has shown no difference in mortality based on the diabetes status, previous control, or complications [1]. Therefore, glycemic control may be important to all COVID-19 patients even if they have no pre-existing diabetes, as most COVID-19 patients are prone to glucose metabolic disorders as a result of stress hyperglycemia and probably the adverse effects of several treatments.

Among the factors that may influence the appearance of stress hyperglycemia in critically ill patients, the release of inflammatory mediators might be one of the physiopathological pathways. Inflammatory cytokines excessively produce glucose by the liver and insulin resistance at the peripheral tissues [8], and the resultant hyperglycemia further exacerbates the inflammatory and oxidative stress response [9]. The modulation of the immune response in patients receiving insulin treatment may partially explain a reduction in morbidity and mortality [24]. In agreement, we have shown that circulating IL-12 concentrations correlated with hyperglycemia at ICU admission and that those patients with stress hyperglycemia showed higher ICU stays.

The amount or glucose delivered through artificial nutrition might be another associated factor with stress hyperglycemia. It has been shown that among the metabolic complications of parenteral nutrition, hyperglycemia is one of them, and this may be especially important in older individuals [7]. Therefore, nutrition-support regimens need to minimize stress hyperglycemia and assist glucose management [25]. Recent recommendations from the European Society of Clinical Nutrition and Metabolism (ESPEN) state that in critically COVID-19 patients who do not tolerate full-dose EN during the first week in the ICU, initiating PN should be weighed on a case-by-case basis [14]. Conversely, in the last two decades, evidence-based recommendations suggest PN use in patients in whom EN cannot be initiated within 24 h of ICU admission or injury [26,27,28], as it produces similar outcomes as EN alone. Even a combination of PN and EN in critically ill patients has been recently recommended as a better approach [29].

However, there is still a paucity of data in critically ill COVID-19 patients regarding the recommendation of the type of artificial nutrition and also its composition. Our results showed that the amount of glucose delivered by artificial nutrition did not differ between COVID-19 and non-COVID-19 patients. It is true that the former needed higher rates of insulin infusion, but this could be related to the use of glucocorticoids as part of their treatment. In the present study, we have also shown that the amount of glucose delivered through parenteral nutrition was associated with higher glycemia at ICU independently of COVID-19 diagnosis.

An important limitation of our study is that it was enabled to find differences in ICU stay and serum osteocalcin but not in the composition of artificial nutrition. In this regard, we showed after performing a multivariate analysis that both ICU stay and osteocalcin were independently associated with COVID-19 diagnosis and the presence of stress hyperglycemia. However, the latter were also associated with longer artificial nutrition support and higher energy needs, so the design of our study precludes us from reaching valid, nonbiased associations regarding artificial nutrition. Therefore, future studies are needed to address the role of the type and composition of artificial nutrition in critically ill COVID-19 patients.

## 5. Conclusions

Hyperglycemia in critically ill COVID-19 patients was associated with longer ICU stays and with higher amounts of glucose delivered through artificial nutrition in both COVID-19 and non-COVID-19 patients. Low circulating osteocalcin was a good marker for stress hyperglycemia.

## Figures and Tables

**Figure 1 nutrients-13-03010-f001:**
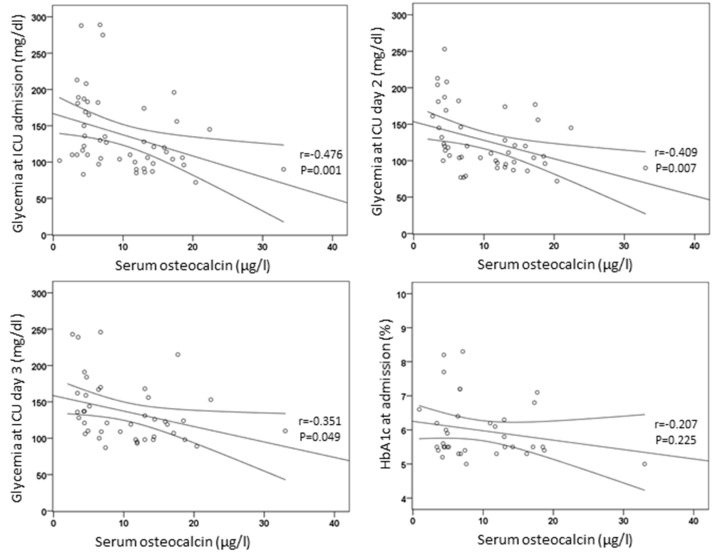
Correlations of circulating osteocalcin with glycemia and HbA1c.

**Figure 2 nutrients-13-03010-f002:**
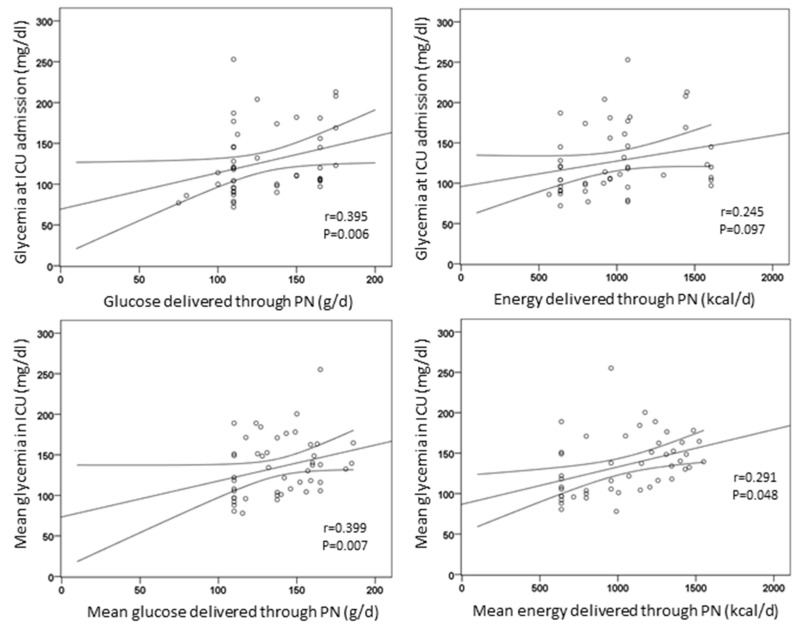
Correlations of energy and glucose delivered through artificial nutrition with glycemia.

**Figure 3 nutrients-13-03010-f003:**
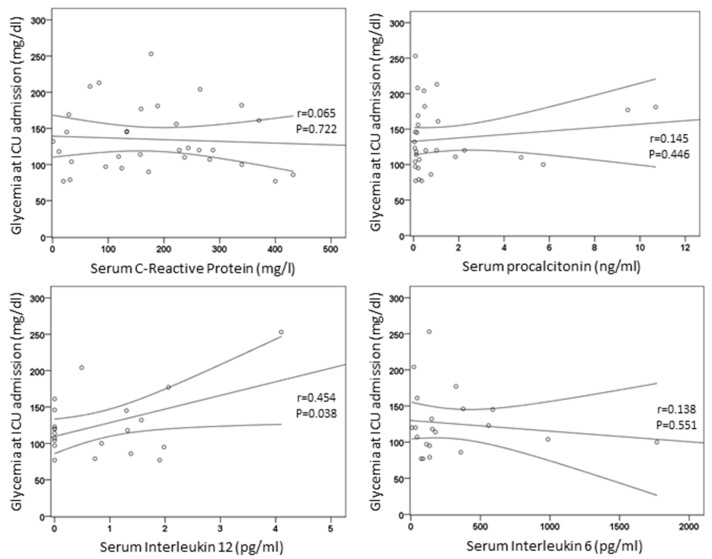
Correlations of serum inflammatory markers with glycemia.

**Table 1 nutrients-13-03010-t001:** Characteristics of the critically ill included patients.

	Patients with COVID-19 (*n* = 23)	Non-COVID-19 Patients (*n* = 26)	*p*
Male sex (*n*, %)	19 (83)	14 (54)	0.039
Age (y)	64 ± 9	71 ± 8	0.005
Time to ICU admission (days)	8.1 ± 12.2	4.0 ± 6.7	0.158
*Glucose metabolism*			
Previous diabetes mellitus (*n*, %)	6 (23)	6 (26)	0.806
HbA1c (%)	6.0 ± 0.9	6.0 ± 0.8	0.977
Glycemia at ICU admission (mg/dl)	148 ± 62	129 ± 41	0.207
Mean glycemia 1 week at ICU (mg/dl)	136 ± 37	128 ± 39	0.523
Patients with stress hyperglycemia (*n*, %) *	10 (48)	12 (46)	0.920
Patients with insulin infusion therapy (*n*, %)	14 (61)	7 (27)	0.022
*Artificial nutrition*			
Time on TPN (days)	15.7 ± 9.6	7.2 ± 10.1	0.004
Time on EN (days)	10.0 ± 12.9	2.1 ± 4.3	0.010
Mean energy delivered (kcal/day)	1222 ± 180	900 ± 329	<0.001
Mean glucose delivered (g/day)	141 ± 15	137 ± 29	0.600
*Metabolic and inflammatory markers*			
Osteocalcin (µg/L)	7.0 ± 3.5	12.9 ± 7.0	<0.001
Creatinine (mg/dl)	1.0 ± 0.6	1.3 ± 1.3	0.309
CRP (mg/L)	181 ± 129	161 ± 105	0.652
Procalcitonin (ng/mL)	1.0 ± 2.2	2.4 ± 3.5	0.156
D-dimer (µg/mL)	3791 ± 5403	-	-
IL-6 (pg/mL)	280 ± 400	-	-
IL-12 (pg/mL)	0.8 ± 1.1	-	-
*Prognostic parameters*			
ICU stay (days)	24 ± 16	9 ± 13	<0.001
Total hospital stay (days)	46 ± 19	32 ± 39	0.138
Mortality (*n*, %)	8 (35)	7 (27)	0.551

Data are means ± SD unless otherwise stated. TPN, total parenteral nutrition; EN, enteral nutrition; CRP, C-reactive protein; IL, interleukin; ICU, intensive care unit. * Stress hyperglycemia was defined as glycemia ≥ 140 mg/dl.

**Table 2 nutrients-13-03010-t002:** Characteristics of patients with and without stress hyperglycemia *.

	Patients with Stress Hyperglycemia (*n* = 22)	Patients without Stress Hyperglycemia (*n* = 27)	*p*
COVID-19 diagnosis (*n*, %)	10 (46)	13 (48)	0.851
Male sex (*n*, %)	14 (64)	19 (70)	0.617
Age (y)	69 ± 9	67 ± 10	0.574
*Glucose metabolism*			
Previous diabetes mellitus (*n*, %)	9 (41)	3 (11)	0.022
HbA1c (%)	6.4 ± 0.9	5.7 ± 0.7	0.013
Glycemia at ICU admission (mg/dl)	162 ± 44	118 ± 42	0.003
Mean glycemia 1-week at ICU (mg/dl)	161 ± 29	112 ± 26	<0.001
*Artificial nutrition*			
Time on TPN (days)	14.7 ± 11.8	8.3 ± 8.9	0.035
Time on EN (days)	8.6 ± 12.6	3.5 ± 6.8	0.075
Mean energy delivered (kcal/day)	1185 ± 296	941 ± 287	0.006
Mean glucose delivered (g/day)	147 ± 24	132 ± 20	0.026
*Metabolic and inflammatory markers*			
Osteocalcin (µg/L)	8.2 ± 5.3	12.3 ± 7.0	0.034
CRP (mg/L)	170 ± 102	180 ± 144	0.808
Procalcitonin (ng/mL)	1.8 ± 3.2	0.8 ± 1.5	0.314
*Prognostic parameters*			
ICU stay (days)	23 ± 17	11 ± 13	0.007
Total hospital stay (days)	47 ± 39	31 ± 23	0.090
Mortality (*n*, %)	5 (19)	10 (45)	0.085

Data are means ± SD unless otherwise stated. TPN, total parenteral nutrition; EN, enteral nutrition; CRP, C-reactive protein; ICU, intensive care unit. * Stress hyperglycemia was defined as glycemia ≥ 140 mg/dl.

**Table 3 nutrients-13-03010-t003:** Prognosis and serum markers of patients with and without COVID-19 and with or without stress hyperglycemia *,^†^.

	COVID-19 with Stress Hyperglycemia (*n* = 10)	COVID-19 without Stress Hyperglycemia (*n* = 13)	Non-COVID-19 with Stress Hyperglycemia (*n* = 12)	Non-COVID-19 without Stress Hyperglycemia (*n* = 14)	*p* for COVID-19 Effect	*p* for Hyperglycemia	*p* for Interaction
Male sex (*n*, %)	9 (90)	10 (77)	5 (42)	9 (64)	0.031	0.678	
Age (y)	65 ± 8	63 ± 10	72 ± 9	71 ± 8	0.006	0.595	0.878
ICU stay (days)	30 ± 17	20 ± 14	17 ± 15	2 ± 2	<0.001	0.002	0.172
Total hospital stay (days)	47 ± 20	46 ± 19	47 ± 52	20 ± 20	0.263	0.110	0.255
Mortality (*n*, %)	5 (50)	3 (23)	5 (42)	2 (14)	0.391	0.120	
Osteocalcin (µg/L)	6.2 ± 4.2	6.9 ± 4.3	9.8 ± 6.3	15.7 ± 6.2	0.001	0.027	0.172
CRP (mg/L)	192 ± 98	173 ± 153	146 ± 108	230 ± 82	0.926	0.560	0.354
Procalcitonin (ng/mL)	1.2 ± 2.9	0.7 ± 1.5	2.4 ± 3.7	1.3 ± 0.7	0.373	0.816	0.904
Patients with insulin infusion (*n*, %)	8 (80)	6 (46)	7 (58)	0 (0)	0.040	0.041	
Time on TPN (days)	16.4 ± 11.5	15.2 ± 8.4	13.3 ± 12.5	2.0 ± 1.3	0.003	0.020	0.060
Time on EN (days)	13.5 ± 17.0	7.2 ± 8.3	4.6 ± 5.5	1.0 ± 0.5	0.004	0.045	0.750
Mean energy delivered (kcal/day)	1274 ± 135	1182 ± 205	1109 ± 373	717 ± 121	<0.001	0.001	0.029
Mean glucose delivered (g/day)	140 ± 16	141 ± 15	153 ± 29	124 ± 21	0.721	0.029	0.015

Data are means ± SD unless otherwise stated. ICU, intensive care unit; CRP, C-reactive protein. * Stress hyperglycemia was defined as glycemia ≥ 140 mg/dl. ^†^, *p* show the results of two-way ANOVA except for categorical variables that were analyzed by corrected χ^2^ tests.

## Data Availability

Restrictions apply to the availability of data generated or analyzed during this study to preserve patient confidentiality or because they were used under license. The corresponding author will on request detail the restrictions and any conditions under which access to some data may be provided.

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
