# Peer review of "Stress Hyperglycemia and Osteocalcin in COVID-19 Critically Ill Patients on Artificial Nutrition"

_nutrients, 2021, doi:10.3390/nu13093010_

Round 1
Reviewer 1 Report
Dear Authors,
- You shouls not use headings in abstract (see instructions for authors)
- Were non-COVID-19 critically ill patients also tested by RT-PCR
to exclude the presence of SARS-CoV-2 in respiratory specimens? When they were tested ( before surgery or on admission to the ICU? Please add this information to methods section. - What was the average time period from covid diagnosis to ICU admission? Please add this information to the tables
- Please specify Inclusion and Exclusion Criteria for study participants, also for non-Covid patients.
- All COVID-19 patients were on mechanical ventilation + how about non+Covid patients
- Regarding mortality -what was the cause of death in both groups? Did the patients from both groups have any vascular/ brain complications (ie stroke, seizure, myocardial infarction)
- Please add a chapter to discussion regarding the possible vascular effect of hyperglycemia - on circulation -vascular resistance, plasma osmolarity, blood viscosity, and metabolism; also on cerebral blood flow), and discuss this possible effect in Covid patients.
- Please create a figure summarising the possible influence of stress hyperglycemia on Covid-19 patients
Author Response
Dear reviewer,
Thank you for your comments and suggestions.
Our response to the points you raised are the following:
1. You should not use headings in abstract (see instructions for authors)
We have corrected this. Thank you.
2. Were non-COVID-19 critically ill patients also tested by RT-PCR
to exclude the presence of SARS-CoV-2 in respiratory specimens? When they were tested (before surgery or on admission to the ICU? Please add this information to methods section.
They were historical controls before covid-19 pandemia occurred. Added in the revised version.
3. What was the average time period from covid diagnosis to ICU admission? Please add this information to the tables
Added to Table 1.
4. Please specify Inclusion and Exclusion Criteria for study participants, also for non-Covid patients.
Added to Methods section.
5. All COVID-19 patients were on mechanical ventilation + how about non+Covid patients
They were critically ill postsurgical patients also on mechanical ventilation. Added to Methods section.
6. Regarding mortality -what was the cause of death in both groups? Did the patients from both groups have any vascular/ brain complications (ie stroke, seizure, myocardial infarction).
The cause of death has been added to the Results section. Thank you.
7. Please add a chapter to discussion regarding the possible vascular effect of hyperglycemia - on circulation -vascular resistance, plasma osmolarity, blood viscosity, and metabolism; also on cerebral blood flow), and discuss this possible effect in Covid patients.
Although we think this is an important pathophysiological link between hyperglycemia and prognosis, we believe this is far beyond our results. In fact we could not accurately nor precisely show any vascular complications apart from pulmonary embolism in our patients.
8. Please create a figure summarising the possible influence of stress hyperglycemia on Covid-19 patients.
As a compromise with the request of Reviewer 2, we have added a new Table showing data in four separated groups: COVID-19 with and without stress hyperglycemia, and non-COVID-19 with and without stress hyperglycemia. We have added new analyses in a separate subheading in the Results section (Ancillary analyses).
Reviewer 2 Report
This manuscript points out the importance of glucose control in nutrition management of patients with COVID-19. Patients with and without COVID-19 were recruited in the study and their ICU stays, glycemic control and metabolic and inflammatory markers were evaluated. The authors concluded that hyperglycemia in critically ill COVID-19 patients was associated with longer ICU stays and higher amounts of glucose delivered through artificial nutrition. Low circulating osteocalcin was a good marker for stress hyperglycemia.
Comments:
Major comments:
- One of the aims of this manuscript is to explore the association of hyperglycemia with the composition of artificial nutrition. However, only the delivered energy and glucose were discussed.
- The results are difficult to follow, e.g., COVID-19 vs. non-COVID-19 patients and stress hyperglycemia vs. without stress hyperglycemia. It is better to divide the patients into 4 groups, i.e., COVID-19 + stress hyperglycemia, COVID-19 - stress hyperglycemia, non-COVID-19 + stress hyperglycemia, and non-COVID-19 - stress hyperglycemia. And then, the authors can compare all those parameters by a 2-way ANOVA to see the main effect of COVID-19 and stress hyperglycemia and the interaction between COVID-19 and stress hyperglycemia.
- Figures1, 2 and 3 can be combined in a table.
- For patients with and without COVID-19, gender, age, and insulin infusion were significantly different. They also infused with different amounts of artificial nutrition. To determine the relationship of hyperglycemia with osteocalcin and other parameters, a multivariate analysis with covariates and confounding factors is needed. More rigorous statistical methods should be used to analyze the data.
- What was the reason for the COVID-19 patients to have higher amounts of delivered glucose and insulin infusion?
- The authors found that patients with COVID-19 needed more frequently insulin infusion therapy for glycemic control and showed lower osteocalcin concentrations. Is it possible that insulin therapy may counteract with the production of osteocalcin in patients with COVID-19?
Minor comments:
- Title can be more simplified and concluded.
- The abstract format and some of the results needs to be revised.
- Line 95: UCI should be ICU.
- Line 149: Table 2 should not be labeled in the subtitle.
- There are minor grammatic
Author Response
Dear reviewer 2,
Thank you very much for your suggestions and comments. Here you will find our response to the points you raised:
Major comments:
1. One of the aims of this manuscript is to explore the association of hyperglycemia with the composition of artificial nutrition. However, only the delivered energy and glucose were discussed.
The main aim was to study the possible association of stress hyperglycemia in COVID-19 critically ill patients with a worse prognosis compared to non-COVID-19 patients, as stated in the final paragraph or the introduction. Also we performed a sample size analysis for the difference in circulating osteocalcin. The rest were secondary endpoints of the study. We have clarified this in the Methods section.
2. The results are difficult to follow, e.g., COVID-19 vs. non-COVID-19 patients and stress hyperglycemia vs. without stress hyperglycemia. It is better to divide the patients into 4 groups, i.e., COVID-19 + stress hyperglycemia, COVID-19 - stress hyperglycemia, non-COVID-19 + stress hyperglycemia, and non-COVID-19 - stress hyperglycemia. And then, the authors can compare all those parameters by a 2-way ANOVA to see the main effect of COVID-19 and stress hyperglycemia and the interaction between COVID-19 and stress hyperglycemia.
We have added the suggested analyses in a separate subheading in the Results section (Ancillary analyses) and included a new Table 3 with data for the suggested 4 groups. We have performed this analysis for the prognostic variables and circulating osteocalcin, the ones that were the main objectives of the study. However it makes no sense to perform a multivariate analysis in glycemia at any time or mean glycemia, as the group with or without stress hyperglycemia was defined by these variables. As also suggested in point 4, we performed a multivariate analysis by linear regression.
3. Figures 1, 2 and 3 can be combined in a table.
As a compromise with the request of Reviewer 1 in point 8, and this suggestion of creating a table, we have left all figures and added a new Table 3 as mentioned before showing data in four separated groups: COVID-19 with and without stress hyperglycemia, and non-COVID-19 with and without stress hyperglycemia.
4. For patients with and without COVID-19, gender, age, and insulin infusion were significantly different. They also infused with different amounts of artificial nutrition. To determine the relationship of hyperglycemia with osteocalcin and other parameters, a multivariate analysis with covariates and confounding factors is needed. More rigorous statistical methods should be used to analyze the data.
We have added the suggested analysis with multivariate linear regression. Thank you for the suggestions.
5. What was the reason for the COVID-19 patients to have higher amounts of delivered glucose and insulin infusion?
As shown in Table 1, mean glucose delivered by artificial nutrition did not differ between COVID-19 and non-COVID-19 patients. It is true that they needed higher rates of insulin infusion, but this could be related to the use of glucocorticoids as part of their treatment. We have included this hypothesis in the Discussion.
6. The authors found that patients with COVID-19 needed more frequently insulin infusion therapy for glycemic control and showed lower osteocalcin concentrations. Is it possible that insulin therapy may counteract with the production of osteocalcin in patients with COVID-19?
We do not believe in this hypothesis, as insulin infusion was initiated at ICU admission if needed when patients already presented stress hyperglycemia, and osteocalcin was measure at ICU admission. Therefore, our data support that osteocalcin is a good early marker of stress hyperglycemia and the need of higher insulin infusion rates.
Minor comments:
- Title can be more simplified and concluded.
- The abstract format and some of the results needs to be revised.
- Line 95: UCI should be ICU.
- Line 149: Table 2 should not be labeled in the subtitle.
- There are minor grammatic
Thank you, corrected in the revised version.
Round 2
Reviewer 2 Report
- Lines 163-165:
- Are those numbers in the parentheses for the numbers of patients? If they are, please add “n = “ before the numbers.
- The mortality numbers and rates are different from those in Table 2。
- When were the blood samples collected from patients administered with PN and EN? For patients with continuous infusion of PN, the blood sugar levels are usually higher than those with EN.
- Please show the p-values of main factors (COVID-19 and stress hyperglycemia) and the interaction between these 2 factors in Table 3. It is hard to understand the results of 2-way ANOVA without those p-values.
- Point 3.6. Ancillary analysis:
- The results of 2-way ANOVA are not used to explain the associations between parameters and main factors. The description of this part should be revised.
- For the multivariate linear regression analysis, the amounts of TPN, EN and insulin administration need to be added to clarify the effects of nutrition support and insulin infusion.
- The discussion session did not discuss the new results in Table 3 and the multivariate linear regression analysis.
- The title can combine the 2 conclusions in the abstract to make it more specific.
Author Response
Thank you very much for your detailed revision of our manuscript. We truly believe that it has helped to improve it and clarify the presented data.
Here you will find the responses to the points you raised:
1. Lines 163-165:
1.1.Are those numbers in the parentheses for the numbers of patients? If they are, please add “n = “ before the numbers.
Yes, they were absolute numbers, we have added “n=” in the text. Thank you
1.2. The mortality numbers and rates are different from those in Table 2。
The text refers to Table 1, we have added this in parenthesis at the end of the paragraph. Table 2 shows mortality in patients classified as with or without stress hyperglycemia.
2. When were the blood samples collected from patients administered with PN and EN? For patients with continuous infusion of PN, the blood sugar levels are usually higher than those with EN.
Blood samples for the measurements of osteocalcin and inflammatory markers were collected at ICU admission, when all patients were on TPN. We have added this to the Methods section. Then, capillary glycemia was monitored thereafter, independently if they were still on TPN or if they could have been changed to EN. We have included in Table 3 data on artificial nutrition in this revised version of the manuscript.
3. Please show the p-values of main factors (COVID-19 and stress hyperglycemia) and the interaction between these 2 factors in Table 3. It is hard to understand the results of 2-way ANOVA without those p-values.
We have added the P values. Thank you for the suggestion.
4. Point 3.6. Ancillary analysis:
4.1. The results of 2-way ANOVA are not used to explain the associations between parameters and main factors. The description of this part should be revised.
We have tried to explain better the results obtained with this analysis.
4.2. For the multivariate linear regression analysis, the amounts of TPN, EN and insulin administration need to be added to clarify the effects of nutrition support and insulin infusion.
We have added new models with these variables in the multivariate linear regression analysis. However, due to the restriction of the sample size, only up to five independent variables were allowed to enter each model (one per 10 patients at maximum). Furthermore, as both patients with COVID-19 and hyperglycemia needed longer periods of artificial nutrition and higher amounts of energy and glucose, these variables showed collinearity, so could not be introduced at the same time in the models.
5. The discussion session did not discuss the new results in Table 3 and the multivariate linear regression analysis.
We have added this part at the end of the Discussion, also indicating the important limitations of our study.
6. The title can combine the 2 conclusions in the abstract to make it more specific.
We have changed the title to a shorter one: “Stress hyperglycemia and osteocalcin in COVID-19 critically ill patients on artificial nutrition”. We hope it is now suitable.